# Interaction of background Ca$^{2+}$ influx, sarcoplasmic reticulum threshold and heart failure in determining propensity for Ca$^{2+}$ waves in sheep heart

David C. Hutchings[1,2] , George W. P. Madders[1] , Barbara C. Niort[1] , Elizabeth F. Bode[1],
Caitlin A. Waddell[1] , Lori S. Woods[1], Katharine M. Dibb[1] , David A. Eisner[1]
and Andrew W. Trafford[1]

[1] *Unit of Cardiac Physiology, Division of Cardiovascular Sciences, Faculty of Biology Medicine and Health, Manchester Academic Health Science Centre, University of Manchester, Manchester, UK*
[2] *Manchester University NHS Foundation Trust, Manchester, UK*

Edited by: Don Bers & Bjorn Knollmann

Linked articles: This article is highlighted in a Perspective article by Fakuade *et al*. To read this article, visit https://doi.org/10.1113/JP283032.

The peer review history is available in the Supporting Information section of this article (https://doi.org/10.1113/JP282168#support-information-section).

**Abstract**  Ventricular arrhythmias can cause death in heart failure (HF). A trigger is the occurrence of Ca$^{2+}$ waves which activate a Na$^+$-Ca$^{2+}$ exchange (NCX) current, leading to delayed after-depolarisations and triggered action potentials. Waves arise when sarcoplasmic reticulum (SR) Ca$^{2+}$ content reaches a threshold and are commonly induced experimentally by raising external Ca$^{2+}$, although the mechanism by which this causes waves is unclear and was the focus of this study. Intracellular Ca$^{2+}$ was measured in voltage-clamped ventricular myocytes from both control sheep and those subjected to rapid pacing to produce HF. Threshold SR Ca$^{2+}$ content was determined by applying caffeine (10 mM) following a wave and integrating wave and caffeine-induced NCX currents. Raising external Ca$^{2+}$ induced waves in a greater proportion of HF cells than control. The associated increase of SR Ca$^{2+}$ content was smaller in HF due to a lower threshold. Raising external Ca$^{2+}$ had no effect on total influx via the L-type Ca$^{2+}$ current, $I_{Ca-L}$, and increased efflux on NCX. Analysis of sarcolemmal fluxes revealed substantial background Ca$^{2+}$ entry which sustains Ca$^{2+}$ efflux during waves in the steady state. Wave frequency and background Ca$^{2+}$ entry were

**David Hutchings** is an NIHR Academic Clinical Lecturer in Cardiology at the University of Manchester and Manchester University NHS Hospitals. He earned his medical degree from the University of Birmingham, undertook junior doctor rotations in Oxford, then moved to Manchester to train as an academic cardiologist. His PhD was funded by a BHF clinical research training fellowship. David is fascinated by mechanisms underlying the heart rhythm and how these go awry in pro-arrhythmic conditions such as channelopathies and cardiomyopathies. He focuses on Ca$^{2+}$ handling and its manipulation with novel treatments. He is especially motivated by relating lab work to treating patients.

The Journal of Physiology

decreased by $Gd^{3+}$ or the TRPC6 inhibitor BI 749327. These agents also blocked $Mn^{2+}$ entry. Inhibiting connexin hemi-channels, TRPC1/4/5, L-type channels or NCX had no effect on background entry. In conclusion, raising external $Ca^{2+}$ induces waves via a background $Ca^{2+}$ influx through TRPC6 channels. The greater propensity to waves in HF results from increased background entry and decreased threshold SR content.

(Received 4 February 2022; accepted after revision 25 February 2022; first published online 1 March 2022)

**Corresponding author** Andrew W. Trafford: University of Manchester, 3.08 Core Technology Facility, Manchester M13 9NT, UK. Email: andrew.w.trafford@manchester.ac.uk

**Abstract figure legend** Raising external $Ca^{2+}$ (1) leads to a background $Ca^{2+}$ influx via TRPC6 channels (2). This $Ca^{2+}$ is pumped into the sarcoplasmic reticulum via SERCA leading to a rise in SR $Ca^{2+}$ content (3). When SR $Ca^{2+}$ content reaches a threshold, spontaneous $Ca^{2+}$ release leads to propagating $Ca^{2+}$ waves (4). In heart failure, the background $Ca^{2+}$ influx is increased and SR threshold decreased, resulting in a greater propensity to $Ca^{2+}$ waves.

## Key points

- Heart failure is a pro-arrhythmic state and arrhythmias are a major cause of death.
- At the cellular level, $Ca^{2+}$ waves resulting in delayed after-depolarisations are a key trigger of arrhythmias. $Ca^{2+}$ waves arise when the sarcoplasmic reticulum (SR) becomes overloaded with $Ca^{2+}$.
- We investigate the mechanism by which raising external $Ca^{2+}$ causes waves, and how this is modified in heart failure.
- We demonstrate that a novel sarcolemmal background $Ca^{2+}$ influx via the TRPC6 channel is responsible for SR $Ca^{2+}$ overload and $Ca^{2+}$ waves.
- The increased propensity for $Ca^{2+}$ waves in heart failure results from an increase of background influx, and a lower threshold SR content.
- The results of the present study highlight a novel mechanism by which $Ca^{2+}$ waves may arise in heart failure, providing a basis for future work and novel therapeutic targets.

## Introduction

Cardiac contraction is activated by an increase of cytoplasmic $Ca^{2+}$ concentration ($[Ca^{2+}]_i$). The bulk of this $Ca^{2+}$ is provided by release from the sarcoplasmic reticulum (SR) by a mechanism known as calcium induced calcium release (CICR) in which $Ca^{2+}$ entering the cell, via the L-type $Ca^{2+}$ current, produces a local increase of $[Ca^{2+}]_i$ which opens the SR $Ca^{2+}$ release channel (ryanodine receptor, RyR). See Bers (2008) and Eisner *et al*. (2017) for reviews. It is well known that SR $Ca^{2+}$ release can also occur in the absence of triggering L-type $Ca^{2+}$ current leading to abnormal waves of CICR (Wier *et al*. 1987). These waves activate delayed afterdepolarizations and thence ventricular ectopic beats and arrhythmias (Ferrier *et al*. 1973; Rosen *et al*. 1973; Lederer & Tsien, 1976). $Ca^{2+}$ waves and their arrhythmogenic consequences occur more frequently in heart failure (Pogwizd *et al*. 2001). $Ca^{2+}$ waves are initiated when the SR $Ca^{2+}$ content exceeds a threshold level and this can occur in one of two ways. (1) If the threshold is decreased, as occurs when the RyR open probability is increased by mutations, for example in catecholaminergic polymorphic ventricular tachycardia (CPVT) (Jiang *et al*.

2005; Kashimura *et al*. 2010). A decreased threshold may account for the increased propensity for waves in heart failure (Belevych *et al*. 2007; Maxwell *et al*. 2012). (2) Waves and delayed afterdepolarizations can also occur when the myocyte is overloaded with $Ca^{2+}$ such that SR $Ca^{2+}$ content is increased above the threshold level (Díaz *et al*. 1997; Jiang *et al*. 2004) as was first demonstrated for digitalis intoxication (Ferrier *et al*. 1973; for review see Venetucci *et al*. 2008).

A commonly used experimental tool to produce $Ca^{2+}$ overload is to elevate the extracellular $Ca^{2+}$ concentration (Kass *et al*. 1978; Hayashi *et al*. 1994; Cheng *et al*. 1996; Díaz *et al*. 1997; Minamikawa *et al*. 1997; Lukyanenko *et al*. 1999; Yang *et al*. 2007; Wasserstrom *et al*. 2010). It is, however, unclear by what mechanism elevating extracellular $Ca^{2+}$ increases SR $Ca^{2+}$ content to the threshold for waves to develop. One possibility might be increased influx through the L-type $Ca^{2+}$ current. However, the effects of an increase of L-type current on SR content are complicated; loading of the SR by increased influx is opposed by increased release and the net effect is hard to predict (Trafford *et al*. 2001). Another explanation is a decrease of $Ca^{2+}$ efflux on sodium calcium exchange

(NCX) due to the increased driving force against which it must transport. Finally, as recently reviewed (Eisner *et al.* 2020), there are other, as yet inadequately characterized, mechanisms by which $Ca^{2+}$ can enter the cell (Terracciano & MacLeod, 1996; Kupittayanant *et al.* 2006; Hutchings *et al.* 2021), including Trp channels (Camacho Londono *et al.* 2015), and connexin hemi channels (Wang *et al.* 2012; De Smet *et al.* 2021).

The aim of the work in this paper was to characterize the mechanisms by which elevating extracellular $Ca^{2+}$ concentration increases the occurrence of $Ca^{2+}$ waves in sheep ventricular myocytes taken from both control animals and in heart failure. We find that this is associated with elevated SR $Ca^{2+}$ content but this is not a consequence of either increased L-type $Ca^{2+}$ current or decreased NCX but, rather, of background $Ca^{2+}$ entry. The majority of this background entry appears to be via TRPC6 channels. Our findings indicate that arrhythmogenic $Ca^{2+}$ waves are produced more easily in myocytes from heart failure animals due to a combination of a larger background influx and a lower threshold SR $Ca^{2+}$ content.

## Methods

### Ethical approval

All procedures involving the use of animals were performed in accordance with The United Kingdom Animals (Scientific Procedures) Act, 1986 and European Union Directive 2010/63. Institutional approval was obtained from The University of Manchester Animal Welfare and Ethical Review Board. Furthermore, the study accords with the ARRIVE guidelines (Percie du Sert *et al.* 2020).

### Induction of heart failure

Female Welsh mountain sheep were group-housed, at $19-21°C$, in a 12:12 h light:dark cycle. Animals had *ad libitum* access to drinking water, and were fed hay and ruminant concentrate. No animals were excluded from the study. Heart failure was induced in 13 adult animals (~18 months age, weight $31.9 \pm 3.7$ kg) via rapid pacing as previously described (Dibb *et al.* 2009; Briston *et al.* 2011; Lawless *et al.* 2019). Briefly, under general anaesthesia (isoflurane $1-4\%$) animals underwent trans-venous insertion of a pacing lead with active fixation to the apex of the right ventricle, and connected to a pacemaker buried subcutaneously in the right pre-scapular position. Subcutaneous Meloxicam ($0.5$ mg kg$^{-1}$) was administered for perioperative analgesia, and Enrofloxacin ($5$ mg kg$^{-1}$) or oxytetracycline ($20$ mg kg$^{-1}$) administered for peri-operative antibiosis. Following a recovery period (at least 7 days) rapid pacing was commenced (210 beats per minute; bpm). Animals were monitored on a daily basis for features of heart failure (cough, dyspnoea). Heart failure animals developed symptoms at $51 \pm 16$ days, at which point they were humanely killed for isolation of cells by anaesthetic overdose (200 mg kg$^{-1}$ intravenous pentobarbitone). Heparin (10,000–25,000 i.u.) was used to prevent coagulation.

### Cellular studies

Left ventricular myocytes were isolated from sheep using a collagenase and protease digestion technique as described previously (Dibb *et al.* 2004; Briston *et al.* 2011).

Voltage clamp was imposed using the whole cell technique. Following rupture of the patch, access resistance (~5 MΩ) was overcome using the switch clamp facility of the Axoclamp-2B voltage clamp amplifier (Axon Instruments, Union City, CA, USA). Electrodes (2–3 MΩ resistance) were filled with a pipette solution containing (in mM): CsCl, 118; MgCl$_2$, 4.0; CaCl$_2$, 0.28; sodium phosphocreatine, 3; HEPES, 10; CsEGTA, 0.02; Na$_2$ATP, 3.1; Na$_2$GTP, 0.42; pH 7.2 with CsOH. For all experiments under voltage clamp, intracellular $Ca^{2+}$ concentration ($[Ca^{2+}]_i$) was measured using the indicator Fura-2 (pentapotassium salt; 100 μM, Invitrogen), loaded via the patch pipette. As indicated in the figure legends, fluorescence was excited either at wavelengths of 365 and 380 nm or 340 and 380 nm and emitted fluorescence detected at $510 \pm 10$ nm. After subtracting background fluorescence, the ratio of light excited at 340 or 365 nm to that excited at 380 nm was used to measure changes in $[Ca^{2+}]_i$.

Cells were held at a holding potential of $-40$ mV and depolarizing pulses to 10 mV applied at 0.5 Hz; L-type $Ca^{2+}$ current and NCX currents were measured as previously described (Trafford *et al.* 1997). Cells were superfused with (in mM): NaCl, 140; KCl, 4.0; MgCl$_2$, 1; HEPES, 10; glucose, 10; CaCl$_2$, 1.8; probenecid, 2; 4-aminopyridine, 5; BaCl$_2$, 0.1; pH 7.34 with NaOH. $Ca^{2+}$ waves were induced by increasing external $Ca^{2+}$ to 10 mM. In some experiments (Fig. 2) it was necessary to elevate external $Ca^{2+}$ to 15 mM to produce waves.

SR content was measured at $-40$ mV by rapidly applying 10 mM caffeine (Sigma-Aldrich, UK) to discharge $Ca^{2+}$ from the SR, and integrating the resulting inward NCX current ($I_{NCX}$) (Varro *et al.* 1993). To determine threshold SR $Ca^{2+}$ content to induce waves, caffeine (10 mM) was added immediately following a wave. The sum of the integrals of the wave and caffeine-induced NCX currents was taken as threshold (Kashimura *et al.* 2010). For all experiments in 1.8 mM external $Ca^{2+}$, total efflux was estimated by multiplying $I_{NCX}$ efflux by a correction factor (1.44) to account for $Ca^{2+}$ removal by PMCA. For experiments in 10 mM external $Ca^{2+}$, no correction factor was used as PMCA

removal is inhibited under these conditions (Bassani *et al.* 1992).

In separate experiments, pharmacological inhibitors were used to examine the identity of the background $Ca^{2+}$ influx in unpatched spontaneously waving cells (Figs 6*Ab* and 7). $[Ca^{2+}]_i$ was measured using the acetoxymethyl ester (AM) form of Fura-4F (Life Technologies, USA). Fluorescence excited at wavelengths of 340 and 380 nm. 18$\beta$-Glycyrrhetinic acid (100 $\mu$M, Sigma, UK) was used to inhibit connnexin hemichannels (Guan *et al.* 1996; Vaiyapuri *et al.* 2012), nicardipine (5 $\mu$M, Stratech, UK) for inhibition of L-type $Ca^{2+}$ channels (Sun *et al.* 1999), BI 749327 (100 nM, MedChem Express, USA) for inhibition of TRPC6 channels (Lin *et al.* 2019), and Pico145 (13 nM, Generon, UK) for inhibition of TRPC1/4/5 channels (Rubaiy *et al.* 2017). Gadolinium chloride (100 $\mu$M, Bio-Techne Ltd, UK) was used as a non-specific inhibitor of background influx (Kupittayanant *et al.* 2006). Inhibitors were dissolved in DMSO (final concentration not exceeding 0.1% v/v) with the exception of gadolinium (dissolved in water). For each experiment, inhibitors were paired with a vehicle control of the same volume.

Finally, to further investigate the background influx, manganese ($Mn^{2+}$) quench was performed, as previously described (Camacho Londono *et al.* 2015). Experiments were performed in $Ca^{2+}$-free superfusion solution. Myocytes were AM-loaded with Fura-4 and excited at near-isosbestic wavelength (365 nm). Initial control recordings showed a slow decline in the $F_{365}$ signal related to a combination of photobleaching and indicator loss. $MnCl_2$ (1 mM, Sigma, UK) was then rapidly applied, leading to $Mn^{2+}$ entry via background $Ca^{2+}$ channels and quenching of the Fura signal. The rate by which $Mn^{2+}$ quenches Fura provides a measure of the rate of $Mn^{2+}$ entry via background channels. Quench rates were determined after subtracting the rate of photobleaching/indicator loss. The rate of quench was normalized to that from in a cell from the same animal in the absence of inhibitors. The effect of the inhibitors 18$\beta$-glycyrrhetinic acid, Pico145, BI 749327, and gadolinium on the quench rate were determined. In some experiments, the rate of decline in the presence of inhibitors was slower than the prior control rate, resulting in apparent negative quench rate. This is probably because of the control rate being exponential rather than linear. In such cases the rate was assigned a value of zero for calculations

All cellular experiments were performed at 37°C.

## Statistics

Data are presented as means $\pm$ standard deviation for *n* cells/*N* animals. As in previous work (Caldwell *et al.* 2014), when comparisons were made between control and HF animals and multiple cells studied from the same animal, linear mixed modelling (SPSS Statistics, IBM, USA) was performed thus accounting for the nested (clustered) design of the experiment (Eisner, 2021). Data was log10 transformed before linear mixed modelling to achieve a normal distribution (Keene, 1995). Categorical variables were compared between groups using the Fischer's exact or chi-squared tests as appropriate. For $Mn^{2+}$ quench experiments, the rate of quench in the presence of a putative inhibitor was paired with that in the absence of inhibitor in a cell from the same isolation using a Wilcoxon matched-pairs signed rank test. Exact *P* values are stated if $P > 0.0001$.

## Results

### Effects of elevation of external $Ca^{2+}$ concentration on $Ca^{2+}$ cycling

Previous work has shown that, in this sheep tachy-pacing model, heart failure decreases the amplitude of the systolic $Ca^{2+}$ transient (Briston *et al.* 2011; Lawless *et al.* 2019). Figure 1*A* shows that increasing external $Ca^{2+}$ concentration from 1.8 to 10 mM increased the amplitude in both control (*a*) and heart failure (*b*) cells, although the percentage increase was greater in heart failure (141 $\pm$ 166%; mean $\pm$ SD) than control (37 $\pm$ 43%; mean $\pm$ SD) (Fig. 1*B*). In heart failure (but not control) cells, raising external $Ca^{2+}$ increased diastolic (*C*) and average (*D*) $[Ca^{2+}]_i$. Figure 1*E* shows that in control cells in 1.8 mM $Ca^{2+}$, only a small proportion of cells showed $Ca^{2+}$ waves and this fraction increased in 10 mM $Ca^{2+}$ to 50%. The propensity to wave was greater in heart failure (78%, $P = 0.03$). Subsequent experiments were designed to investigate the role of SR $Ca^{2+}$ content in this increased incidence of waves in elevated external $Ca^{2+}$.

### Effects of external $Ca^{2+}$ concentration on sarcoplasmic reticulum $Ca^{2+}$ content

In the experiments illustrated in Fig. 2*A*, the SR $Ca^{2+}$ content was measured from the integral of the caffeine-evoked NCX current. Waves were absent in 1.8 mM external $Ca^{2+}$ in both the control (*a*) and heart failure cells (*b*), but appeared in 10 mM. This was accompanied by an increase of SR $Ca^{2+}$ content as shown by the integral of the caffeine-evoked NCX current. The summary data of Fig. 2*B* (black points) show the measurements for all those control cells which were induced to wave by elevation of external $Ca^{2+}$ concentration. The appearance of waves was associated with an increase of SR $Ca^{2+}$ content from 42 $\pm$ 48 to 128 $\pm$ 61 $\mu$mol $l^{-1}$ (mean $\pm$ SD). Figure 2*Ab* and the mean data of Fig. 2*B* demonstrate that, in heart failure,

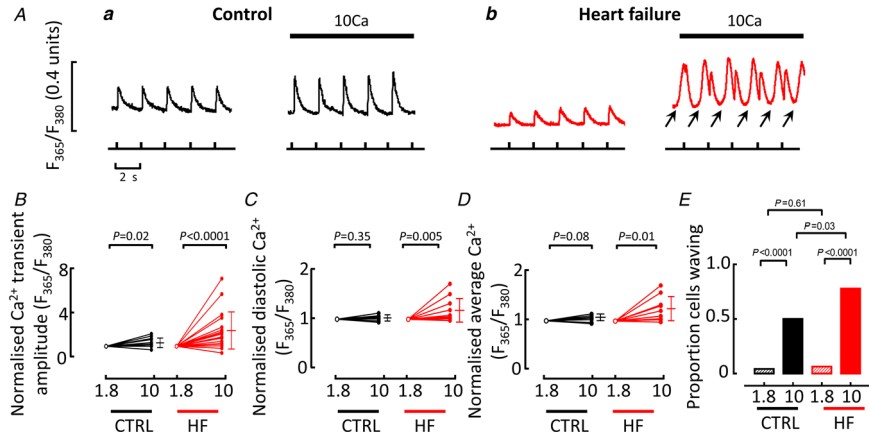

**Figure 1. Effects of increasing external Ca²⁺ concentration in control and heart failure cells**
*A*, specimen records showing effects of elevating Ca²⁺ from 1.8 to 10 mM in a control (*a*) and heart failure (HF) (*b*) myocytes. In this and subsequent figures, cells were stimulated with 100 ms duration pulses from a holding potential of −40 to 10 mV, applied at 0.5 Hz. Arrows indicate Ca²⁺ waves. *B*–*D*, summary data (normalized to 1.8 mM Ca²⁺) showing the effects of increasing external Ca²⁺ on Ca²⁺ transient amplitude (*B*), diastolic $[Ca^{2+}]_i$ (*C*) and average $[Ca^{2+}]_i$ (*D*). *E*, summary data of the proportion of cells showing waves. Mean ± SD shown to the right of data in 10 mM Ca²⁺. For Ca²⁺ transient amplitude; control 11 cells/8 animals one sample *t* test, HF 19 cells/8 animals Wilcoxon matched pairs signed rank test. For diastolic $[Ca^{2+}]_i$; control 9 cells/6 animals one sample *t* test, HF 12 cells/7 animals Wilcoxon matched pairs signed rank test. For average Ca²⁺; control 8 cells/5 animals, HF 11 cells/7 animals, one sample *t* test for both comparisons. For proportion of cells waving; control 1.8 mM Ca²⁺ 117 cells/41 animals, control 10 mM Ca²⁺ 30 cells/16 animals, HF 1.8 mM Ca²⁺ 31 cells/10 animals, HF 10 mM Ca²⁺ 27 cells/10 animals, chi-squared test for all comparisons. [Colour figure can be viewed at wileyonlinelibrary.com]

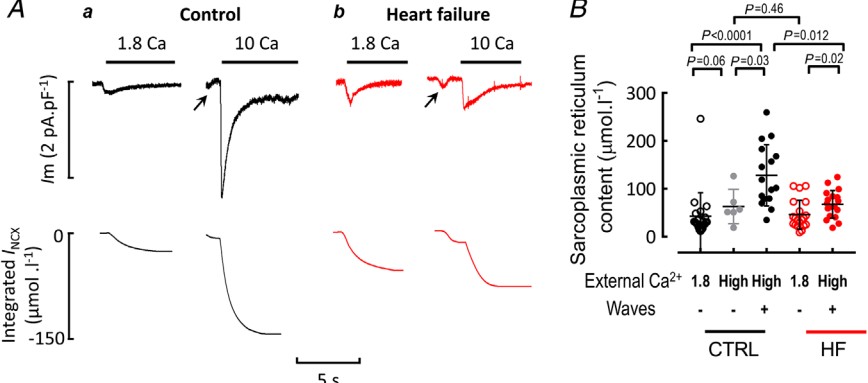

**Figure 2. Effects of external Ca²⁺ concentration on SR Ca²⁺ content and threshold for waves**
*A*, original data. Traces show: top, membrane current; bottom, integral of current. Records are taken from representative examples from control (*a*) and HF (*b*) myocytes. In both, the left-hand traces were recorded in 1.8 mM Ca²⁺ and the right-hand in 10 mM. 10 mM caffeine was applied for the period shown by the horizontal bars. Arrows show inward currents produced by Ca²⁺ waves. *B*, summary data. In this, and subsequent diagrams, error bars denote ± SD for both control and heart failure, the left-hand points (open symbols) show SR Ca²⁺ content measured in 1.8 mM Ca²⁺ (in the absence of waves: 21 cells from 15 animals in control and 20 cells from 7 animals in HF). The right-hand points (+waves) show the SR Ca²⁺ content in those cells which displayed waves in elevated Ca²⁺. This was achieved in 9 control cells (from 7 animals) and 18 HF cells (from 9 animals) by elevating external Ca²⁺ to 10 mM, and in 7 control cells (from 4 animals) by elevating external Ca²⁺ to 15 mM. Cells from control animals which did not display waves in high Ca²⁺ are also shown (grey symbols, marked '−waves', total 6 cells from 5 animals; two of which were in 15 mM Ca²⁺ and 4 in 10 mM Ca²⁺). For control 1.8 Ca *vs*. control high Ca '−waves', Mann-Whitney test. For control 1.8 Ca *vs*. control high Ca '+waves', Mann-Whitney test. For control high Ca '+waves' *vs*. control high Ca '−waves', unpaired *t* test. For control high Ca '+waves' *vs*. HF high Ca '+waves', mixed effects linear mixed modelling. For HF 1.8 Ca *vs*. HF high Ca, Mann-Whitney test. [Colour figure can be viewed at wileyonlinelibrary.com]

the induction of waves by elevation of external $Ca^{2+}$ was associated with a smaller increase of SR $Ca^{2+}$ than was the case for control cells; SR $Ca^{2+}$ content increased from $45 \pm 29$ to $67 \pm 28$ $\mu$mol l$^{-1}$ (mean $\pm$ SD). The lower SR $Ca^{2+}$ content in heart failure indicates that the threshold for $Ca^{2+}$ waves is lower in heart failure cells than control. This explains why SR $Ca^{2+}$ content rises less in heart failure as it is limited by the production of $Ca^{2+}$ waves. To further illustrate this, control cells which were below threshold and not displaying waves in high $Ca^{2+}$ are shown as grey points in the summary data. These cells had lower SR contents than control cells with waves, but similar SR contents to HF cells with waves.

The difference of threshold for production of $Ca^{2+}$ waves provides one explanation as to why heart failure cells are more likely to exhibit $Ca^{2+}$ waves. We have, however, argued previously that a difference of threshold by itself is insufficient to produce waves (Venetucci *et al.* 2007). Specifically, something must maintain the SR $Ca^{2+}$

content to balance the extra efflux resulting from $Ca^{2+}$ waves. Subsequent experiments were therefore designed to investigate the change of $Ca^{2+}$ fluxes.

### Effects on the L-type Ca²⁺ current

Figure 3 addresses the question as to whether the increases of SR content and wave probability produced by elevating external $Ca^{2+}$ involve changes in the L-type $Ca^{2+}$ current. In control cells, elevating $Ca^{2+}$ increases the peak L-type current (Fig. 3*Aa* and *Cb*). On average, in 10 mM $Ca^{2+}$ in the steady state, the amplitude of the L-type current increased from $4.57 \pm 3.33$ to $6.58 \pm 4.59$ pA·pF$^{-1}$ (mean $\pm$ SD, $P < 0.0001$) but this was not accompanied by any change of total $Ca^{2+}$ entry, assessed from the integral (Fig. 3*B* and *Cc*), as the current inactivates more quickly. In heart failure cells, the increase of wave probability was not associated with changes of either amplitude or integral of the L-type $Ca^{2+}$ current. In 10 mM $Ca^{2+}$, the integral

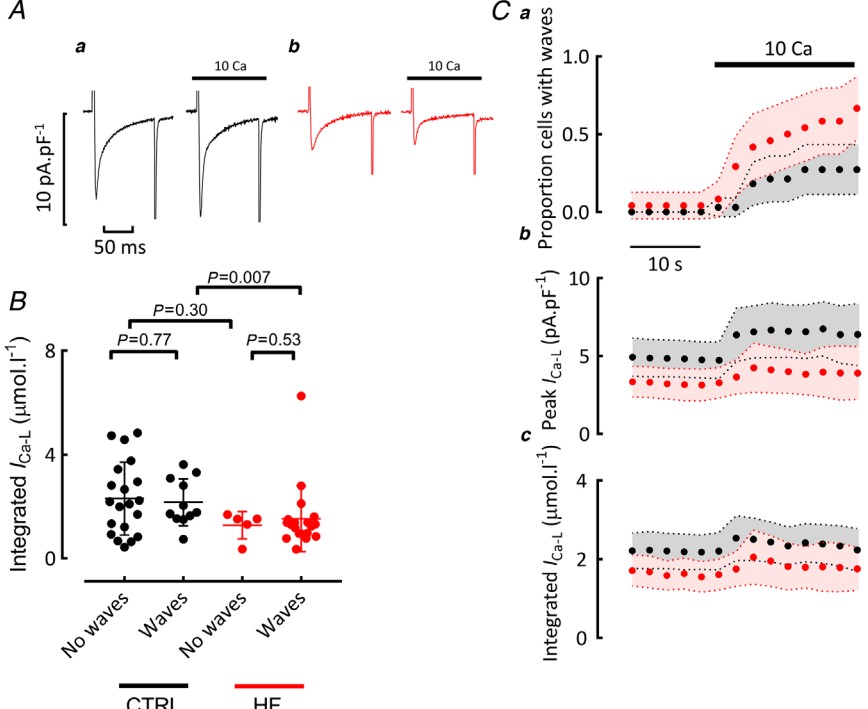

**Figure 3. The effects of external Ca²⁺ concentration on the L-type Ca²⁺ current**
*A*, specimen paired records showing the effects of elevating external $Ca^{2+}$ from 1.8 to 10 mM. In all panels 100 ms duration depolarizing pulses were applied at 0.5 Hz to +10 mV from a holding potential of −40 mV. Panels show: *a*, control; *b*, heart failure. In both panels the left-hand trace was obtained in 1.8 and the right-hand in 10 mM external $Ca^{2+}$ from the same cell. *B*, integral of the L-type $Ca^{2+}$ current in 10 mM external $Ca^{2+}$. Black symbols from control cells, red heart failure. In both, data are separated by whether the cells showed waves or not. Control: no waves 20 cells/14 animals, with waves 11 cells/5 animals. Heart failure: no waves 5 cells/3 animals, with waves 19 cells/8 animals. For control no waves *vs.* with waves, unpaired *t* test. For HF no waves *vs.* with waves, Mann-Whitney test. For control no waves *vs.* HF no waves, mixed effects linear mixed modelling. For control with waves *vs.* HF with waves, Mann-Whitney test. *C*, time course of mean data (31 control and 24 heart failure cells). Graphs show: *a*, fraction of cells displaying waves; *b*, mean peak L-type $Ca^{2+}$ current; *c*, mean integral of L-type current. Black symbols, control; red symbols, heart failure. External $Ca^{2+}$ concentration was increased from 1.8 to 10 mM for the period shown. Shaded areas show 95% confidence limits. [Colour figure can be viewed at wileyonlinelibrary.com]

of the L-type current was lower in heart failure than in control ($P = 0.03$). Thus, influx via the L-type current does not determine whether cells wave in either cell type, and therefore cannot explain the greater propensity to waves in heart failure.

### Effects on Ca²⁺ efflux and background influx

Figure 4 shows steady state measurements of Ca²⁺ efflux recorded from cells exposed to 10 mM external Ca²⁺. In cells without Ca²⁺ waves, such as the control myocyte illustrated in Fig. 4*Aa*, the NCX current was observed as a 'tail' during the decay of $[Ca^{2+}]_i$ on repolarization. As shown in Fig. 4*B*, in both control and heart failure, increasing external Ca²⁺ increased the NCX tail efflux to similar levels. Therefore, decreased NCX activity cannot be the explanation of the increase of SR Ca²⁺ load. When waves were present (Fig. 4*Ab* and *c*), the NCX tail current was followed by an NCX current activated by the Ca²⁺ wave. Mean data for the wave-associated Ca²⁺ efflux are shown in Fig. 4*C*. There is a considerable spread of values, with those at the zero level representing cells which did not have waves. The average wave-associated efflux is greater in heart failure than control because a greater fraction of cells displays Ca²⁺ waves. The time course of changes of Ca²⁺ efflux produced by elevating external Ca²⁺ is shown in Fig. 5*A*. Elevation of external Ca²⁺ increases Ca²⁺ efflux via NCX on both the tail (*a*) and waves (*b*). Figure 5*Ac* (filled symbols) plots the sum of these two components of Ca²⁺ efflux. In 1.8 mM this efflux is equal to the influx on the L-type current (open symbols) but greatly exceeds it in 10 mM. In control cells, elevation of external Ca²⁺ increases Ca²⁺ efflux per cycle (2 s) from $2.5 \pm 2.1$ to $8.2 \pm 7.1$ μmol l⁻¹ (mean ± SD) while, in heart failure, the respective values are $2.9 \pm 3.5$ and $10.7 \pm 7.3$ μmol l⁻¹ (both comparisons between the last data points in 1.8 and 10 mM Ca²⁺). In the steady state, total Ca²⁺ efflux must equal influx so the fact that influx through the L-type channel is smaller than the efflux means that there must be another 'background' component of influx. The magnitude of this influx is shown in Fig. 5*B*; on average the background influx is greater in heart failure ($9.3 \pm 6.8$ μmol l⁻¹ per cycle; mean ± SD) than in control ($6.0 \pm 7.2$). The analysis of Fig. 5*C* measures the background influx as a function, not only of cell type but, in addition, whether the cells are showing Ca²⁺ waves or not. Analysis of those cells without waves in 10 mM Ca²⁺ shows no significant difference in background influx between control and heart failure. In both control and heart failure, those cells that show waves have a greater background influx than those that do not and, finally, the background influx is similar in control ($13.2 \pm 6.5$ μmol l⁻¹ per cycle; mean ± SD) and heart failure ($10.8 \pm 6.8$) cells that wave. A background Ca²⁺ influx has been demonstrated previously but its identity was unknown (Kupittayanant *et al.* 2006). The remainder of the experiments were therefore designed to characterize this flux.

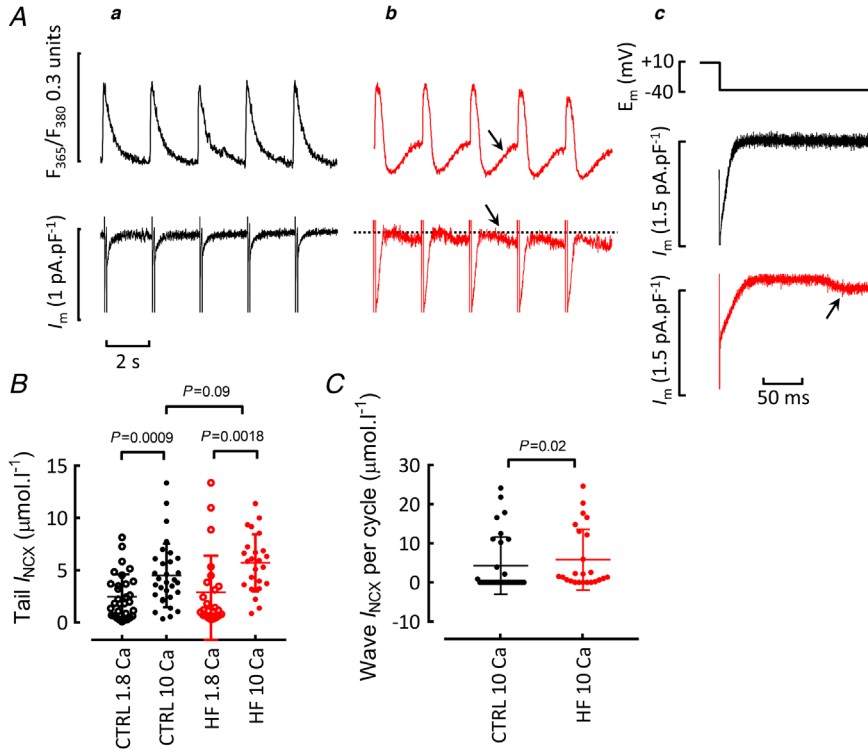

**Figure 4. Measurement of Ca²⁺ efflux in elevated external Ca²⁺**

*A*, original records: top, $[Ca^{2+}]_i$, bottom, membrane current. *a*, control; *b*, heart failure; *c*, expanded, averaged (five sweeps) membrane current records. All data obtained in 10 mM external Ca²⁺. Arrows denote a Ca²⁺ wave and accompanying inward current. *B*, average Ca²⁺ efflux on NCX during the Ca²⁺ transient (tail). Data shown from both 1.8 and 10 mM external Ca²⁺ in control and HF. Control: 31 cells/18 animals, HF 24 cells/9 animals. For comparisons between 1.8 and 10 mM Ca²⁺ (in both HF and control), paired *t* tests. For comparison between HF and control, mixed effects linear mixed modelling. *C*, average Ca²⁺ efflux on NCX per cycle during waves. Control: 31 cells/18 animals, HF 24 cells/9 animals, Mann-Whitney test. [Colour figure can be viewed at wileyonlinelibrary.com]

## Identity of the background influx

We first examined whether $Ca^{2+}$ could be entering between waves by NCX acting in reverse. $Ni^{2+}$ was used to inhibit NCX (Kimura *et al.* 1987). Initial experiments under voltage clamp confirmed $Ni^{2+}$ (10 mM) blocked $I_{NCX}$ in 15 mM Ca (reversible loss of wave $I_{NCX}$ current in three cells exposed to $Ni^{2+}$, Fig. 6*Aa*). $Ni^{2+}$ also blocked $I_{Ca-L}$. Further experiments were performed in unpatched cells pre-exposed to $Ni^{2+}$ for at least 30 s. Raising external $Ca^{2+}$ to 15 mM in the presence of $Ni^{2+}$ induced waves in a similar proportion of cells to control (waving in $Ni^{2+}$ 52.0% *vs.* Ctrl 59.7%, $P = 0.51$). $Ni^{2+}$ increased the frequency of waves (Fig. 6*Ab*). In 12 cells which did not wave in $Ni^{2+}$, washing $Ni^{2+}$ out did not induce waves in any cells (not shown).

Figure 6*B* shows an alternative method of assessing the contribution of NCX to the background influx. The record in Fig. 6*Ba* shows the typical rise in $[Ca^{2+}]_i$ when 15 mM $Ca^{2+}$ solution was applied to a cell which had been pre-exposed to a $Ca^{2+}$-free solution. To block NCX completely, the cell in Fig. 6*Bb* was pre-exposed to $Ca^{2+}$ and $Na^+$-free solution ($Na^+$ replaced by $Li^+$); here a similar rise in $[Ca^{2+}]_i$ is seen, indicating the background $Ca^{2+}$ entry is not via reverse-mode NCX (see also summary data Fig. 6*Bc*).

To investigate other possible candidates for the background influx, specific inhibitors were tested in cells displaying spontaneous waves in high $Ca^{2+}$ (15 mM; Fig. 7). Under these conditions, a decrease of background influx should decrease wave frequency. Gadolinium ($Gd^{3+}$, 100 $\mu$M) and the TRPC6 inhibitor, BI 749327 (100 nM) both reduced the frequency of waves (Fig. 7*Aa* and *b*, and *Ba* and *b*). In contrast, block of $I_{Ca-L}$ with nicardipine (5 $\mu$M) had no effect on wave frequency (Fig. 7*Bc*). Neither the application of the TRPC1/4/5 channel inhibitor Pico145 nor pre-incubating cells with $\beta$-glycrrhetinic acid had any effect on waves (Fig. 7*Bd* and *e*).

Subsequent experiments were designed to measure background $Ca^{2+}$ influx more directly using the quench of $Ca^{2+}$-sensitive indicators produced by $Mn^{2+}$ (Camacho Londono *et al.* 2015). Figure 8*A* shows typical quenches of the Fura-2 signal when $Mn^{2+}$ was applied. The quench was suppressed in the presence of $Gd^{3+}$ and BI 749327; see also Fig. 8*Ba* and *b*. In contrast, exposing cells to Pico145 (to inhibit TRPC 1/4/5 channels) or $\beta$-glycrrhetinic acid (to inhibit connexin hemichannels) had no effect on $Mn^{2+}$ quench rates (Fig. 8*Bc* and *d*).

The above findings suggest the background $Ca^{2+}$ entry responsible for $Ca^{2+}$ waves is independent of $I_{Ca-L}$ and

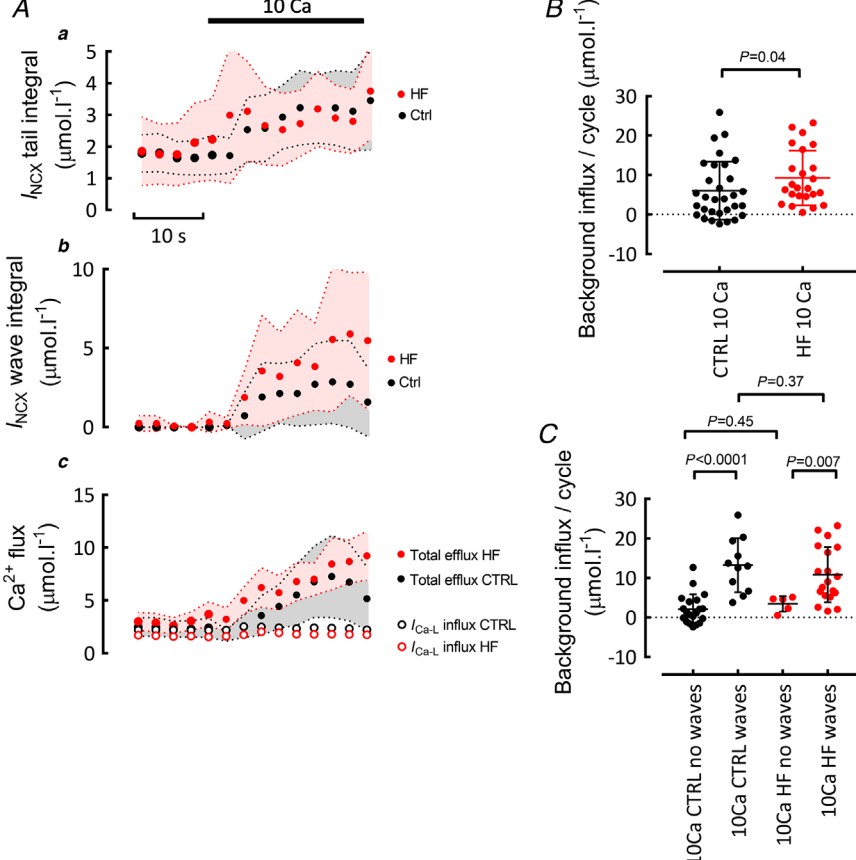

**Figure 5. Estimation of background flux**
*A*, time course of mean data showing the effects of elevating external $Ca^{2+}$ from 1.8 to 10 mM: *a*, efflux during $Ca^{2+}$ transient (tail); *b*, efflux on waves; *c*, total efflux (filled symbols) compared with $Ca^{2+}$ influx on L-type current (open symbols). Shaded areas show 95% confidence limits. *B*, background influx in the steady state in 10 mM external $Ca^{2+}$ in control (left) and heart failure (right). Control 31 cells/18 animals, HF 24 cells/9 animals, mixed effects linear mixed modelling. *C*, background influx as a function of both whether waves are present and cell type. Control: no waves 20 cells/14 animals, with waves 11 cells/5 animals. HF: no waves 5 cells/4 animals, with waves 19 cells/9 animals. For control, no waves *vs.* with waves, Mann-Whitney test. For HF, no waves *vs.* with waves, Mann-Whitney test. For comparisons between HF and control, mixed effects linear mixed modelling. [Colour figure can be viewed at wileyonlinelibrary.com]

$I_{NCX}$, is Gd sensitive, and appears to be carried via TRPC6 channels, with no apparent role for TRPC1/4/5 channels or connexin hemichannels.

## Discussion

The main results of this paper are that, in elevated external Ca$^{2+}$, ventricular myocytes from sheep with heart failure are more likely to show Ca$^{2+}$ waves than are those from control animals. Two factors are responsible for this: (i) lower SR Ca$^{2+}$ threshold for waves in heart failure; and (ii) a larger proportion of HF cells have a high background Ca$^{2+}$ influx. Finally, we have shown that Ca$^{2+}$ entry through TRPC6 channels is the most likely candidate for the molecular nature of this background Ca$^{2+}$ entry.

### SR Ca$^{2+}$ threshold

We find that the SR Ca$^{2+}$ threshold at which waves occur in heart failure is about half of the value observed in control cells, a result which is qualitatively in agreement with previous work (Belevych *et al.* 2007; Maxwell *et al.* 2012) and may result from RyR dysfunction and an increase of RyR open probability, as a consequence of factors such as phosphorylation (Marx *et al.* 2000; Ai

*et al.* 2005; van Oort *et al.* 2010), oxidation (Terentyev *et al.* 2008), and decreased S-nitrosylation (Gonzalez *et al.* 2007). This lower threshold and the consequent Ca$^{2+}$ release from the SR during waves explain why elevation of external Ca$^{2+}$ concentration increases SR Ca$^{2+}$ content less in heart failure cells than in control. It is worth noting that whilst SR Ca$^{2+}$ content is the same in control and heart failure in 1.8 mM Ca$^{2+}$, the lower threshold SR Ca$^{2+}$ content in heart failure means that the normal SR Ca$^{2+}$ content is closer to threshold, and thus one potential explanation for the greater propensity for Ca$^{2+}$ waves. While the decrease of threshold in heart failure is important, it cannot be the only explanation for the greater occurrence of waves in heart failure. This is because, in the steady state, the Ca$^{2+}$ efflux on waves must be balanced by additional Ca$^{2+}$ influx (Venetucci *et al.* 2007; Eisner *et al.* 2013). It is therefore important to consider Ca$^{2+}$ influx and, more generally, Ca$^{2+}$ flux balance.

### Ca$^{2+}$ flux balance

**L-type Ca$^{2+}$ channel and NCX.** In control cells we find that an increase of external Ca$^{2+}$ concentration produced a small increase of the amplitude of the L-type current. It is unclear why no similar effect was seen in heart failure. In neither control nor heart failure, however, was there

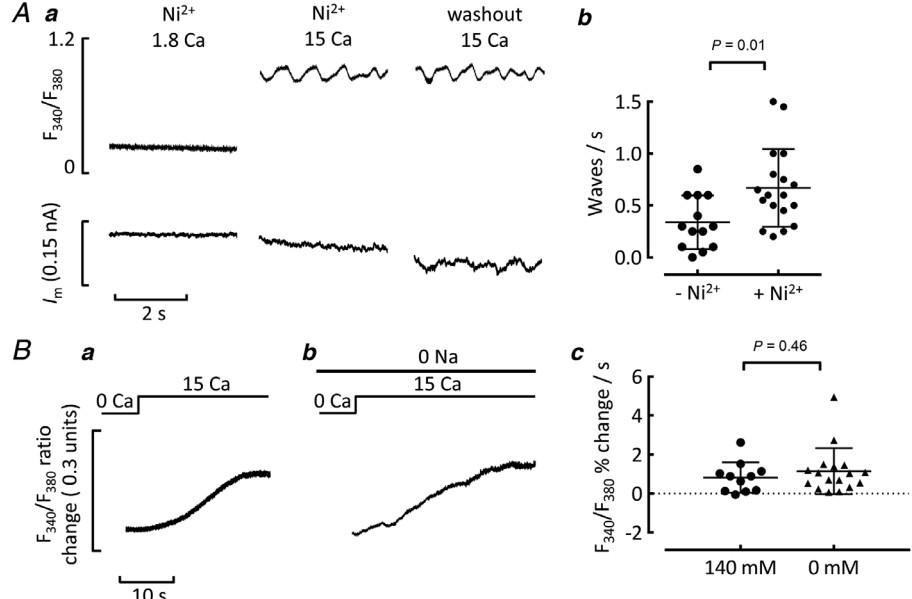

**Figure 6. Effect of NCX block on Ca$^{2+}$ waves and background Ca$^{2+}$ entry**
*Aa*, representative recordings from a single cell under voltage clamp. Panels show (from left to right): Ni$^{2+}$ (10 mM) and 1.8 mM Ca$^{2+}$; Ni$^{2+}$ and 15 mM Ca$^{2+}$; Ni$^{2+}$ washout in 15 mM Ca$^{2+}$. *Ab*, summary data for effects of Ni$^{2+}$ (10 mM) on waves in unpatched cells in 10 mM Ca$^{2+}$. Unpaired data. *B*, representative increases in [Ca$^{2+}$]$_i$ when cells were exposed to 15 mM external Ca$^{2+}$. The cells had been in Ca$^{2+}$-free solution for at least 2 min before raising Ca$^{2+}$, and caffeine (10 mM) was present throughout to prevent Ca$^{2+}$ uptake into the SR. Records show: *a*, control; *b*, Na-free (a different cell). *c*, summary data for the maximum rate of rise. For *Ab*, n = 13 cells/4 animals Ctrl *vs.* n = 18 cells/3 animals Ni$^{2+}$, unpaired *t* test. For *Bc*, n = 11 cells/3 animals in 140 mM Na$^+$, n = 17 cells/3 animals in 0 mM Na$^+$, Mann-Whitney test.

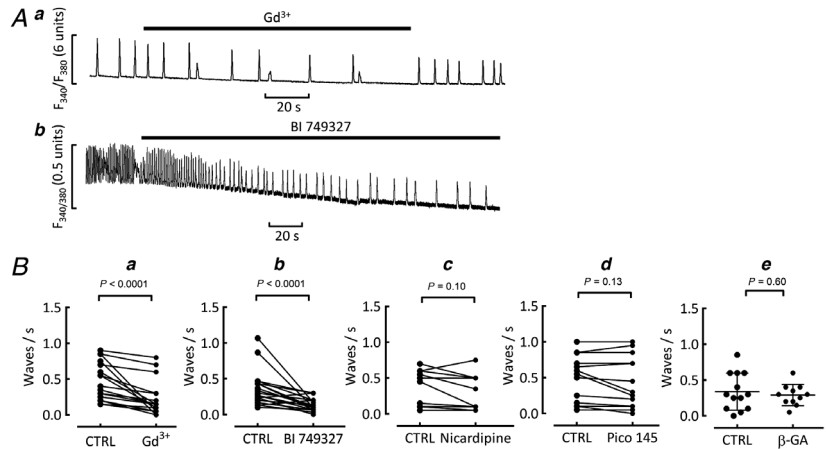

**Figure 7. Effect of inhibitors on spontaneous Ca²⁺ waves**

Waves were induced in control cells by raising external $Ca^{2+}$ to 15 mM. *A*, representative $Ca^{2+}$ recordings in waving cells exposed to $Gd^{3+}$ and its washout (*a*) and BI 749327 (*b*). *B*, mean effects of inhibitors on wave frequency: *a*, mean effect of Gd, paired data from *n* = 18 cells/3 animals. Wilcoxon matched-pairs signed rank test; *b*, mean effect of BI 749327, Wilcoxon matched-pairs signed rank test on paired data from *n* = 20 cells/4 animals; *c*, mean effect of nicardipine, paired *t* test from *n* = 11 cells/3 animals; *d*, mean effect of Pico145 on wave frequency, paired *t* test from *n* = 13 cells/3 animals; *e*, mean effect of pre-incubation with *β*-glycrrhetinic acid on wave frequency, unpaired *t* test from *n* = 13 cells/4 animals (control) and 11 cells/3 animals (*β*-GA).

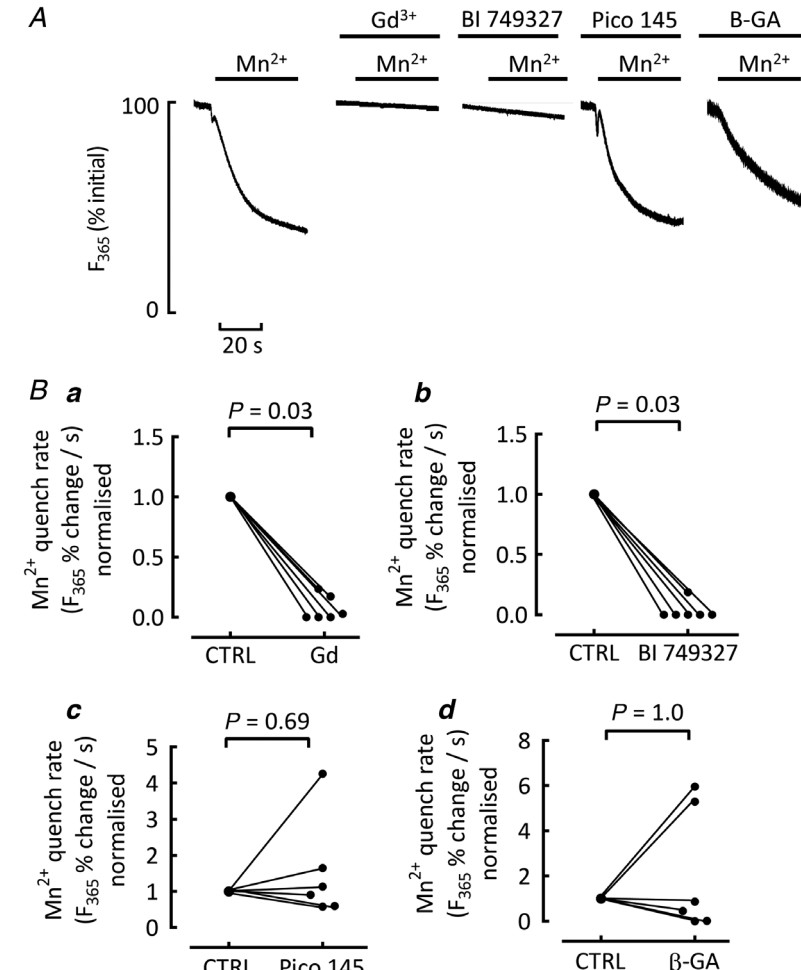

**Figure 8. Assessment of background influx with Mn²⁺ quench**

*A*, representative recordings of Fura signal quench ($F_{365}$) in single cells exposed to $Mn^{2+}$ (1 mM). The effects of $Mn^{2+}$ were tested in control cells (left), and when exposed to gadolinium, BI 749327, Pico 145 and *β*-glycrrhetinic acid. *B*, summary mean data. For analysis, cells were randomly paired with control cells from the same animal and the rate of quench normalized to the control value. For each inhibitor *Ba–d*, *n* = 6 cells/3 animals, Wilcoxon ranked pairs signed rank test.

any effect on the amount of $Ca^{2+}$ entering through this current as assessed from its integral. It appears that the increase of amplitude is balanced by faster inactivation. This lack of effect on $Ca^{2+}$ entry via the L-type current contrasts with the marked increase of efflux on NCX, primarily due to that activated by $Ca^{2+}$ waves. As shown in Fig. 5, in 1.8 mM $Ca^{2+}$, the NCX efflux balances influx on the L-type current. In contrast, in 10 mM $Ca^{2+}$, efflux is 3.9 times influx though the L-type current. We conclude, therefore, that the increase of SR $Ca^{2+}$ content cannot be due to either an increase of $Ca^{2+}$ entry on the L-type current nor a decrease of efflux on NCX.

**Background $Ca^{2+}$ entry.** Given that the cell must be in a steady state, there must be an additional component of $Ca^{2+}$ influx to balance the increased efflux. We have estimated this background influx from the difference between measured influx and efflux in the steady state. Two points about this calculation needs addressing. (i) As in previous work, our electrophysiological approach measures $Ca^{2+}$ efflux on the electrogenic NCX but not that on the electroneutral PMCA. It is likely that 10 mM external $Ca^{2+}$ inhibits PMCA (Bassani *et al.* 1995) and we have therefore not corrected for this flux. (ii) The measurements of NCX current are made with respect to the baseline current and therefore ignore any contribution of NCX to this baseline. Both factors mean that our estimation of the background $Ca^{2+}$ influx is, if anything, an underestimate. The data show that, not only is the background influx increased by elevation of external $Ca^{2+}$ concentration, but (Fig. 5*B*), on average, it is greater in heart failure than control cells. More strikingly, the magnitude of this background influx determines whether or not waves occur. As shown in Fig. 5*C*, in control cells, those that show $Ca^{2+}$ waves have a higher background influx than those that do not. This larger background influx in cells with $Ca^{2+}$ waves is also apparent in heart failure cells. In summary, the magnitude of the background influx appears to be the single factor that is most correlated with whether or not waves occur and accounts for the bulk of the difference between control and heart failure.

Previous work on ventricular myocytes has shown that the refilling of an empty SR requires $Ca^{2+}$ influx from outside the cell and a component of this occurs by a mechanism which does not involve either the L-type $Ca^{2+}$ current or NCX (Terracciano & MacLeod, 1996). A background $Ca^{2+}$ entry pathway which is increased by hyperpolarization and is sensitive to $Gd^{3+}$ has been identified (Kupittayanant *et al.* 2006). The fact that $Ca^{2+}$ waves can be produced even when the cell membrane potential is held constant and there is no $Ca^{2+}$ entry through the L-type channel (Díaz *et al.* 1997) also argues for substantial background influx. In keeping with this previous work, our findings in spontaneously waving cells

treated with nicardipine showed no role for $Ca^{2+}$ entry via $I_{Ca-L}$ in generating waves, while waves were reduced by $Gd^{3+}$. The effect of BI 749327 in both suppressing waves and reducing the background influx in $Mn^{2+}$ quench experiments is strongly suggestive of a role for TRPC6 in mediating this background influx. In contrast, there was no evidence for other candidates such as connexin hemichannels or TRPC1/4/5 contrasting with the role for TRPC1/4 in promoting background $Ca^{2+}$ entry in mouse ventricle (Camacho Londono *et al.* 2015).

## Study limitations

It has to be noted that, with the exception of $Mn^{2+}$-quench experiments, the background $Ca^{2+}$ entry was studied under conditions of elevated $Ca^{2+}$. While relevant to much other experimental work, it is unclear to what extent this background entry contributes under normal conditions. A major limitation to our understanding thus far has been the lack of knowledge of the identity of this background $Ca^{2+}$ entry. As reviewed recently (Eisner *et al.* 2020), several candidates were proposed including $Ca^{2+}$ entry through TRP channels (Camacho Londono *et al.* 2015) or connexin hemichannels (Wang *et al.* 2012). Our use of specific inhibitors has found a role for TRPC6 channels. We have not, however, examined other members of the TRP family. It is noteworthy that TRPC channel expression increases in heart failure (Bush *et al.* 2006; Kuwahara *et al.* 2006; Morine *et al.* 2016), and future work should address the role of specific mechanisms of these channels in generating the background influx under physiological conditions, as well as their contribution to adverse cardiac remodelling (Gao *et al.* 2012; Camacho Londono *et al.* 2015).

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

# Additional information

## Data availability statement

The data that support the findings of this study are available from the corresponding author upon reasonable request.

## Competing interests

No competing interests declared.

## Author contributions

Cellular experiments (D.C.H., B.C.N.); animal model (G.W.P.M., D.C.H., C.A.W., L.S.W.); experimental concepts, direction (D.C.H., D.A.E., A.W.T., K.M.D., E.F.B.); manuscript preparation (D.A.E., D.C.H., A.W.T.); funding (A.W.T., K.M.D., D.A.E.). All authors approved the final version of the manuscript. All authors revised the manuscript critically for important intellectual content. All experiments were performed at The University of Manchester. All persons designated as authors qualify for authorship, and all those who qualify for authorship are listed.

## Funding

The work was supported by grants from the British Heart Foundation: FS/15/28/31476, FS/12/57/29717, FS/09/036/27823, FS/20/6/34990, CH/2000004/12801, AA/18/4/34221 and Medical Research Council: MR/K501211/1. D.C.H. was supported by a clinical lectureship from the NIHR.

## Keywords

$Ca^{2+}$, heart failure, sarcoplasmic reticulum, threshold, waves

## Supporting information

Additional supporting information can be found online in the Supporting Information section at the end of the HTML view of the article. Supporting information files available:

**Statistical Summary Document**
**Peer Review History**

