## [Peer Review History · The Journal of Physiology]

Interaction of background Ca²⁺ influx, sarcoplasmic reticulum threshold and heart failure in determining propensity for Ca²⁺ waves in sheep heart

David Hutchings, George Madders, Barbara Niort, Elizabeth Bode, Caitlin Waddell, Lori Woods, Katharine Dibb, David A Eisner, and Andrew W Trafford

DOI: 10.1113/JP282168

Corresponding author(s): Andrew Trafford (andrew.w.trafford@manchester.ac.uk)

Review Timeline:

Submission Date:	06-May-2020
Editorial Decision:	26-May-2020
Resubmission Received:	04-Feb-2022
Editorial Decision:	10-Feb-2022
Revision Received:	21-Feb-2022
Accepted:	24-Feb-2022

Senior Editor: Don Bers

Reviewing Editor: Bjorn Knollmann

Transaction Report:

Dear Professor Trafford,

Re: JP-RP-2020-280079 "Interaction of background Ca²⁺ influx, sarcoplasmic reticulum threshold and heart failure in determining propensity for Ca²⁺ waves in sheep heart" by David Hutchings, George Madders, Elizabeth Bode, Caitlin Waddell, Lori Woods, Katharine Dibb, David A Eisner, and Andrew W Trafford

Thank you for submitting your manuscript to The Journal of Physiology. It has been assessed by a Reviewing Editor and by 2 Referees and the reports are copied below.

Please let your co-authors know of the following editorial decision as quickly as possible.

As you will see, in its current form, the manuscript is not acceptable for publication in The Journal of Physiology. In comments to me, the Reviewing Editor expressed interest in the potential of this study, but much work still needs to be done (and this may include new experiments) in order to satisfactorily address the concerns raised in the reports.

In view of this interest, I would like to offer you the opportunity to carry out all of the changes requested in full, and to resubmit a new manuscript using the "Submit Special Case Resubmission for JP-RP-2020-280079..." on your homepage.

We cannot, of course, guarantee ultimate acceptance at this stage, and do not encourage resubmission if the authors are unwilling or feel unable to satisfactorily address the concerns that have been raised.

A new manuscript would be renumbered and redated, but the original referees would be consulted wherever possible. An additional referee's opinion could be sought, if the Reviewing Editor felt it necessary. A full response to each of the reports should be uploaded with a new version.

I hope that the points raised in the reports will be helpful to you.

Yours sincerely,

Professor Don M. Bers
Senior Editor
The Journal of Physiology
<https://jp.msubmit.net>
<http://jp.physoc.org>
The Physiological Society
Hodgkin Huxley House
30 Farringdon Lane
London, EC1R 3AW
UK
<http://www.physoc.org>
<http://journals.physoc.org>

EDITOR COMMENTS

Both expert reviewers found merit in the experimental approach and results, and I concur. But the current conclusions that there is an additional background Ca conductance are merely based on finding a lack of difference in NCX and LTCC. As commented on also by the reviewers, the authors need to provide direct experimental evidence for the mechanism of the background conductance.

REFEREE COMMENTS

Referee #1:

The paper by Hutchings and colleagues employs sophisticated methodology to evaluate specific calcium alterations in ventricular myocytes obtained from a well characterized heart failure sheep model. They describe changes in ventricular calcium handling that suggests the participation of other calcium "transport" mechanisms besides the well-defined calcium influx routes in the predisposition of heart failure patients to ventricular arrhythmias. The findings are novel and study is well conducted as it provides a new direction for further study.

Specific Comments:

1. The authors suggest that the background influx causes calcium overload in the HF myocytes leading to the production of waves. And the production of waves explains why SR Ca²⁺ content rises less in heart failure with 10 mM external Ca²⁺. If so, was the SR calcium content in the subset of myocytes that did not develop waves with 10 mM Ca²⁺ determined? If yes, this could be shown to further demonstrate the difference in SR calcium content in the absence of Ca²⁺ waves, thus adding more strength to the above-mentioned argument.
2. Considering the SR Ca²⁺ threshold is reduced in HF, do the authors think an intermediate concentration of external Ca²⁺ would suffice to induce waves in HF? Experiments with intermediate concentrations would further demonstrate the propensity of HF to develop waves. This might be informative since a significant amount of CTRL myocytes developed waves with 10 mM Ca²⁺.
3. The authors suggest that increased susceptibility to Ca²⁺ waves may be due to increased SR Ca²⁺ content. In addition, increased leak may be due to RyR2 dysfunction. This should be discussed as well.
4. SERCA activity has been shown to be reduced in HF (Briston et al. 2011) in this model and was suggested to be balanced out by increased sarcolemma efflux via NCX. However, here the authors found comparable tail currents. This potential discrepancy should be discussed.

Minor Comments.

1. Consistency in the use of Ca²⁺ instead of calcium e.g. Figure 2 legend.

Referee #2:

The authors explore new mechanisms of cardiac disease related Ca²⁺ mishandling using ventricular myocytes from ovine large animal model of heart failure. Using whole cell voltage clamp and Ca²⁺ imaging they demonstrate that application of supra-physiological 10 mM Ca²⁺ results in higher frequency of Ca²⁺ waves in myocytes from failing hearts. Given that differences between L-type Ca²⁺ mediated influx were not detected while Na⁺/Ca²⁺ mediated outflux increased 4 times, the authors conclude that myocytes possess background Ca²⁺ current putatively via TRP or connexin hemichannels to balance the fluxes. This current is proposed being the key for Ca²⁺ waves generation and is higher in heart failure. My comments are as follows:

Major

- 1) The manuscript will greatly benefit from testing pharmacological inhibitors for connexin hemichannels 18-beta-glycyrrhetic acid or mefloquine, as well as TRPC channels blockers. It will significantly strengthen conclusions.
- 2) The authors use indirect approach to estimate background Ca²⁺ influx. It would be beneficial to compare results by measuring influx more directly using Mn quenching (Camacho Londono et al 2015).

Minor

- 3) There is a good chance that background LTCC-mediated influx is increased in HF at -40 mV holding potential because presumably in HF the channels are phosphorylated. This can provide an alternative explanation why background Ca²⁺ influx is seen as higher in failing myocytes.
- 4) Could you please clarify why NCX-mediated outflux cannot be reduced between pulses in the absence of waves at 10 mM extracellular Ca²⁺ vs 1.8?
- 5) Do you think that background Ca²⁺ influx will not be affected by intracellular Na⁺ known to be higher in heart failure?

ADDITIONAL FORMATTING REQUIREMENTS:

-Author photo and profile. First (or joint first) authors are asked to provide a short biography (no more than 100 words for one author or 150 words in total for joint first authors) and a portrait photograph. These should be uploaded and clearly labelled with the revised version of the manuscript. See Information for Authors for further details.

-You must start the Methods section with a paragraph headed Ethical Approval. A detailed explanation of journal policy and regulations on animal experimentation is given in Principles and standards for reporting animal experiments in The Journal of Physiology and Experimental Physiology by David Grundy J Physiol, 593: 2547-2549. doi:10.1113/JP270818. Authors should confirm in their Methods section that their experiments were carried out according to the guidelines laid down by their institution's animal welfare committee, and conform to the principles and regulations as described in the Editorial by Grundy (2015). The Methods section must contain details of the anaesthetic regime: anaesthetic used, dose and route of administration and method of killing the experimental animals.

-The Reference List must be in Journal format

-Your manuscript must include a complete Additional Information section

- The Journal of Physiology funds authors of provisionally accepted papers to use the premium BioRender site to create high resolution schematic figures. Follow this link and enter your details and the manuscript number to create and download figures. Upload these as the figure files for your revised submission. If you choose not to take up this offer we require figures to be of similar quality and resolution. If you are opting out of this service to authors, state this in the Comments section on the Detailed Information page of the submission form.

-Please upload separate high-quality figure files via the submission form.

-A Statistical Summary Document, summarising the statistics presented in the manuscript, is required upon revision. Authors must comply with our policy on statistics, data and graphical presentation. Revised manuscripts that do not conform to The Journal's policy will be returned to authors. Please carefully read the requirements here: https://jp.msubmit.net/cgi-bin/main.plex?form_type=display_requirements#statistics

-A Data Availability Statement is required for all papers reporting original data, both in the manuscript and on the submission form. All data supporting the results in the paper should be archived in an appropriate public repository and the Statement should describe the availability or the absence of shared data. Authors must include in their Statement a link to the repository they have used, reference the data in the appropriate sections(s) of their manuscript and cite the data they have shared in the References section. Whenever possible the scripts and other artefacts used to generate the analyses presented in the paper should also be publicly archived. If sharing data compromises ethical standards or legal requirements then authors are not expected to share it, but should note this in their Statement. Authors may wish to use the Standard Templates for Author Use to select appropriate wording. For more information, see our Statistics Policy.

Confidential Review

06-May-2020

Hutchings et al (Interaction of background Ca²⁺ influx.....). Response to reviewers

Our detailed responses to the comments of the reviewers and editors are below (in red). We have performed additional experiments to more directly measure the background Ca²⁺ influx, and establish its molecular identity. Using manganese quench experiments and studying waving cells, we find that the majority of this Ca²⁺ influx is sensitive to Gd³⁺ and appears to be carried through TRPC6 channels.

We believe we have addressed all of the reviewer's comments in full.

Dear Professor Trafford,

Re: JP-RP-2020-280079 "Interaction of background Ca²⁺ influx, sarcoplasmic reticulum threshold and heart failure in determining propensity for Ca²⁺ waves in sheep heart" by David Hutchings, George Madders, Elizabeth Bode, Caitlin Waddell, Lori Woods, Katharine Dibb, David A Eisner, and Andrew W Trafford

Thank you for submitting your manuscript to The Journal of Physiology. It has been assessed by a Reviewing Editor and by 2 Referees and the reports are copied below.

Please let your co-authors know of the following editorial decision as quickly as possible.

As you will see, in its current form, the manuscript is not acceptable for publication in The Journal of Physiology. In comments to me, the Reviewing Editor expressed interest in the potential of this study, but much work still needs to be done (and this may include new experiments) in order to satisfactorily address the concerns raised in the reports.

In view of this interest, I would like to offer you the opportunity to carry out all of the changes requested in full, and to resubmit a new manuscript using the "Submit Special Case Resubmission for JP-RP-2020-280079..." on your homepage.

We cannot, of course, guarantee ultimate acceptance at this stage, and do not encourage resubmission if the authors are unwilling or feel unable to satisfactorily address the concerns that have been raised.

A new manuscript would be renumbered and redated, but the original referees would be consulted wherever possible. An additional referee's opinion could be sought, if the Reviewing

Editor felt it necessary. A full response to each of the reports should be uploaded with a new version.

I hope that the points raised in the reports will be helpful to you.

Yours sincerely,

Professor Don M. Bers

Senior Editor

EDITOR COMMENTS

Both expert reviewers found merit in the experimental approach and results, and i concur. But the current conclusions that there is an additinal background Ca conductance are merely based on finding a lack of difference in NCX and LTCC. As commented on also by the reviewers, the authors need to provide direct experimental evidence for the mechanism of the background conductance.

Thank you. We have performed additional experiments to more directly measure the background Ca²⁺ influx, and establish its molecular identity. Using manganese quench experiments and studying waving cells, we find that the majority of this Ca²⁺ influx is sensitive to Gd³⁺ and appears to be carried through TRPC6 channels.

We believe we have addressed all of the reviewers comments in full.

REFEREE COMMENTS

Referee #1:

The paper by Hutchings and colleagues employs sophisticated methodology to evaluate specific calcium alterations in ventricular myocytes obtained from a well characterized heart failure sheep

model. They describe changes in ventricular calcium handling that suggests the participation of other calcium "transport" mechanisms besides the well-defined calcium influx routes in the predisposition of heart failure patients to ventricular arrhythmias. The findings are novel and study is well conducted as it provides a new direction for further study.

Specific Comments:

1. The authors suggest that the background influx causes calcium overload in the HF myocytes leading to the production of waves. And the production of waves explains why SR Ca²⁺ content rises less in heart failure with 10 mM external Ca²⁺. If so, was the SR calcium content in the subset of myocytes that did not develop waves with 10 mM Ca²⁺ determined? If yes, this could be shown to further demonstrate the difference in SR calcium content in the absence of Ca²⁺ waves, thus adding more strength to the above-mentioned argument.

We have measured the SR content in six control cells in high Ca²⁺ which did not show waves and therefore were below threshold (mean 63 μmol/l, for comparison, SR content was 42 μmol/l in 1.8 Ca²⁺). This is substantially lower than the threshold SR content for control cells (128 μmol/l), but similar to the threshold SR content observed in HF (67 μmol/l). As such these findings support our contention that there is a threshold SR content in determining waves, and that this is reduced in HF, thus preventing SR content from rising further in HF cells.

We are grateful to the reviewer for this comment. We now show the summary data for this in the histogram in Fig 2B and refer to it in the text (bottom of page 10).

In HF cells, because the majority of cells displayed waves in high Ca²⁺, it was impossible to measure a SR content below threshold.

2. Considering the SR Ca²⁺ threshold is reduced in HF, do the authors think an intermediate concentration of external Ca²⁺ would suffice to induce waves in HF? Experiments with intermediate concentrations would further demonstrate the propensity of HF to develop waves. This might be informative since a significant amount of CTRL myocytes developed waves with 10 mM Ca²⁺.

During preliminary experiments, lower concentrations of external Ca²⁺ were tested in control cells (5 and 7.5 mM). Unfortunately, this was not reliable in eliciting Ca²⁺ waves in these cells, with waves observed in only a minority. This would make it difficult to compare threshold SR content between HF and control.

3. The authors suggest that increased susceptibility to Ca²⁺ waves may be due to increased SR Ca²⁺ content. In addition, increased leak may be due to RyR2 dysfunction. This should be discussed as well.

We now emphasise this point in the discussion (page 14 para 2 'SR Ca²⁺ threshold').

4. SERCA activity has been shown to be reduced in HF (Briston et al. 2011) in this model and was suggested to be balanced out by increased sarcolemma efflux via NCX. However, here the authors found comparable tail currents. This potential discrepancy should be discussed.

Two factors affect the NCX tail current in HF. The smaller Ca^{2+} transient will decrease it, however this will be offset by the increase of NCX (Briston et al) thereby providing an explanation for the similarity of the tail current in control and HF.

Minor Comments.

1. Consistency in the use of Ca^{2+} instead of calcium e.g. Figure 2 legend.

This has been amended here and elsewhere in the text.

Referee #2:

The authors explore new mechanisms of cardiac disease related Ca^{2+} mishandling using ventricular myocytes from ovine large animal model of heart failure. Using whole cell voltage clamp and Ca^{2+} imaging they demonstrate that application of supraphysiological 10 mM Ca^{2+} results in higher frequency of Ca^{2+} waves in myocytes from failing hearts. Given that differences between L-type Ca^{2+} mediated influx were not detected while $\text{Na}^{+}/\text{Ca}^{2+}$ mediated outflux increased 4 times, the authors conclude that myocytes possess background Ca^{2+} current putatively via TRP or connexin hemichannels to balance the fluxes. This current is proposed being the key for Ca^{2+} waves generation and is higher in heart failure. My comments are as follows:

Major

1) The manuscript will greatly benefit from testing pharmacological inhibitors for connexin hemichannels 18-beta-glycyrrhetic acid or mefloquine, as well as TRPC channels blockers. It will significantly strengthen conclusions.

We thank the reviewer for this suggestion and now describe our findings with specific inhibitors (BI 749327 to inhibit TRPC 6, Pico 145 to inhibit TRPC 1/4/5, 18β-glycyrrhetic acid to inhibit connexin hemi channels, nifedipine to inhibit I_{Ca-L} , Ni^{2+} and Na-free solutions to block reverse-mode NCX, and the non-specific inhibitor Gadolinium), in the revised manuscript. We also test their effects on manganese quench rates (see below). Neither Pico145 or 18β-GA had any effect on Ca^{2+} waves or on manganese quench rate. There was no effect of nifedipine on waves, while Ni^{2+} increased wave frequency. In contrast the background influx was both Gadolinium and BI-749327-sensitive, accordingly demonstrating a role for TRPC6.

2) The authors use indirect approach to estimate background Ca^{2+} influx. It would be beneficial to compare results by measuring influx more directly using Mn quenching (Camacho Londono et al 2015).

We thank the reviewer for this suggestion and have performed Mn quench experiments (Figure 8). We found no effect with a TRPC1/4/5 channel blocker or a connexin hemichannel blocker on quench rates. In contrast, the quench was suppressed by BI-749327 or Gadolinium. As such, the channels responsible for the background influx are sensitive to both BI-749327 and Gadolinium. This indicates a key role for TRPC6.

Minor

3) There is a good chance that background LTCC-mediated influx is increased in HF at -40 mV holding potential because presumably in HF the channels are phosphorylated. This can provide an alternative explanation why background Ca^{2+} influx is seen as higher in failing myocytes.

We thank the reviewer for the comment. To address this question experimentally we used the I_{Ca-L} inhibitor nifedipine and found that this did not modify Ca^{2+} waves or the propensity to waves (results page 13 para 2, and Figure 7Bc).

4) Could you please clarify why NCX-mediated outflux cannot be reduced between pulses in the absence of waves at 10 mM extracellular Ca^{2+} vs 1.8?

We agree with the reviewer that raising external Ca^{2+} will slow Ca^{2+} removal by NCX. We observed an increase in the NCX tail current, as well as NCX removal via waves, and concluded that raising Ca^{2+} increases NCX efflux. As the reviewer comments, our methodology ignores the contribution which NCX makes to the current baseline between pulses. It is therefore feasible that 10 mM Ca^{2+} may reduce NCX-efflux between beats. We have used the NCX inhibitor Ni^{2+} as well as Na-free solution and find that these have no effect on the background influx, ruling out a role for reverse-mode NCX.

5) Do you think that background Ca²⁺ influx will not be affected by intracellular Na⁺ known to be higher in heart failure?

We assume that the reviewer is suggesting that “reverse mode” NCX may contribute to the background influx and this contribution may be increased in HF due to elevated intracellular Na concentration. It should be noted that our original measurement of the background influx (Kupittayanant et al 2006) was performed in the complete absence of Na⁺.

To address whether intracellular Na⁺ may be contributing to the background influx via NCX acting in reverse, we performed experiments in cells pre-exposed to Ni²⁺ to block I_{NCX} as well as experiments in Na-free solutions. Here we found that block of NCX had no effect on the background influx.

ADDITIONAL FORMATTING REQUIREMENTS:

-Author photo and profile. First (or joint first) authors are asked to provide a short biography (no more than 100 words for one author or 150 words in total for joint first authors) and a portrait photograph. These should be uploaded and clearly labelled with the revised version of the manuscript. See Information for Authors for further details.

-You must start the Methods section with a paragraph headed Ethical Approval. A detailed explanation of journal policy and regulations on animal experimentation is given in Principles and standards for reporting animal experiments in The Journal of Physiology and Experimental Physiology by David Grundy J Physiol, 593: 2547-2549. doi:10.1113/JP270818. Authors should confirm in their Methods section that their experiments were carried out according to the guidelines laid down by their institution's animal welfare committee, and conform to the principles and regulations as described in the Editorial by Grundy (2015). The Methods section must contain details of the anaesthetic regime: anaesthetic used, dose and route of administration and method of killing the experimental animals.

-The Reference List must be in Journal format

-Your manuscript must include a complete Additional Information section

- The Journal of Physiology funds authors of provisionally accepted papers to use the premium BioRender site to create high resolution schematic figures. Follow this link and enter your details and the manuscript number to create and download figures. Upload these as the figure files for your revised submission. If you choose not to take up this offer we require figures to be of similar quality and resolution. If you are opting out of this service to authors, state this in the Comments section on the Detailed Information page of the submission form.

-Please upload separate high-quality figure files via the submission form.

-A Statistical Summary Document, summarising the statistics presented in the manuscript, is required upon revision. Authors must comply with our policy on statistics, data and graphical presentation. Revised manuscripts that do not conform to The Journal's policy will be returned to authors. Please carefully read the requirements here: https://jp.msubmit.net/cgi-bin/main.plex?form_type=display_requirements#statistics

-A Data Availability Statement is required for all papers reporting original data, both in the manuscript and on the submission form. All data supporting the results in the paper should be archived in an appropriate public repository and the Statement should describe the availability or the absence of shared data. Authors must include in their Statement a link to the repository they have used, reference the data in the appropriate section(s) of their manuscript and cite the data they have shared in the References section. Whenever possible the scripts and other artefacts used to generate the analyses presented in the paper should also be publicly archived. If sharing data compromises ethical standards or legal requirements then authors are not expected to share it, but should note this in their Statement. Authors may wish to use the Standard Templates for Author Use to select appropriate wording. For more information, see our Statistics Policy.

PROFESSOR ANDREW W TRAFFORD
Unit of Cardiac Physiology
University of Manchester
3.24 Core Technology Facility
46 Grafton Street
Manchester
M13 9NT

Email: Andrew.W.Trafford@manchester.ac.uk

Phone: +44-161-275-7969
Fax: +44-161-275-2703

Dr Kim Barrett
Editor-in-Chief
The Journal of Physiology

4 February 2022

Dear Dr Barrett,

RE: Original article, JP-RP-2022-282168X 'Interaction of background Ca^{2+} influx and sarcoplasmic reticulum threshold determine the propensity to Ca^{2+} waves and effects of heart failure in sheep heart.'

On behalf of all of the co-authors I would like to thank you for your decision letter of 26th May 2020 and for inviting us to resubmit our article as a 'special case resubmission' for publication in your journal. We are grateful to the editorial team and reviewers for their comments, and believe we have addressed all of these in full.

Specifically, we have gone on to characterise the molecular identity of the background influx responsible for Ca^{2+} waves, and find this is sensitive to both Gd^{3+} and the TRPC6 inhibitor, BI 749327. These agents were effective in both preventing the entry of Mn^{2+} into the cell and suppressing Ca^{2+} waves, and indicate a key role for TRPC6 in mediating the background flux. In contrast, we do not find significant background Ca^{2+} entry via $I_{\text{Ca-L}}$, reverse-mode I_{NCX} , TRPC1/4/5, or connexin hemichannels.

In combination with our finding of a greater background influx in heart failure, we believe these findings represent an important conceptual advance in the genesis of pro-arrhythmic Ca^{2+} waves. We hope that our paper has the requisite originality and scientific rigor to merit publication in the *Journal of Physiology*.

We enclose a marked up copy of the revised manuscript and a response to the reviewers. As a point of note, the paper was originally handled by then Senior Editor Dr Don Bers, but we did not have the option of selecting him as Senior Editor on the submission portal. We have therefore nominated you. Once again, I thank the editorial team and reviewers for their careful reviews, and look forward to receiving your decision in due course.

Yours Sincerely,

Prof. A.W. Trafford
Chair of Cardiac Pathophysiology

Dear Professor Trafford,

Re: JP-RP-2022-282168X "Interaction of background Ca²⁺ influx, sarcoplasmic reticulum threshold and heart failure in determining propensity for Ca²⁺ waves in sheep heart" by David Hutchings, George Madders, Barbara Niort, Elizabeth Bode, Caitlin Waddell, Lori Woods, Katharine Dibb, David A Eisner, and Andrew W Trafford

Thank you for submitting your manuscript to The Journal of Physiology. It has been assessed by a Reviewing Editor and by 2 expert Referees and I am pleased to tell you that it is considered to be acceptable for publication. Before formal acceptance, however, you just need to include an Abstract Figure (and accompanying legend) - please see further details below.

The reports are copied at the end of this email. Please address all of the points and incorporate all requested revisions, or explain in your Response to Referees why a change has not been made.

NEW POLICY: In order to improve the transparency of its peer review process The Journal of Physiology publishes online as supporting information the peer review history of all articles accepted for publication. Readers will have access to decision letters, including all Editors' comments and referee reports, for each version of the manuscript and any author responses to peer review comments. Referees can decide whether or not they wish to be named on the peer review history document.

Authors are asked to use The Journal's premium BioRender (<https://biorender.com/>) account to create/redrawn their Abstract Figures. Information on how to access The Journal's premium BioRender account is here: <https://physoc.onlinelibrary.wiley.com/journal/14697793/biorender-access> and authors are expected to use this service. This will enable Authors to download high-resolution versions of their figures.

I hope you will find the comments helpful and have no difficulty returning your revisions within 2 weeks.

Your revised manuscript should be submitted online using the links in Author Tasks Link Not Available.

Any image files uploaded with the previous version are retained on the system. Please ensure you replace or remove all files that have been revised.

REVISION CHECKLIST:

- Article file, including any tables and figure legends, must be in an editable format (eg Word)
- Abstract figure file (see above)
- Statistical Summary Document
- Upload each figure as a separate high quality file
- Upload a full Response to Referees, including a response to any Senior and Reviewing Editor Comments;
- Upload a copy of the manuscript with the changes highlighted.

- A potential 'Cover Art' file for consideration as the Issue's cover image;
- Appropriate Supporting Information (Video, audio or data set https://jp.msubmit.net/cgi-bin/main.plex?form_type=display_requirements#supp).

To create your 'Response to Referees' copy all the reports, including any comments from the Senior and Reviewing Editors, into a Word, or similar, file and respond to each point in colour or CAPITALS and upload this when you submit your revision.

I look forward to receiving your revised submission.

If you have any queries please reply to this email and staff will be happy to assist.

Yours sincerely,

Professor Don M. Bers
Senior Editor
The Journal of Physiology
<https://jp.msubmit.net>
<http://jp.physoc.org>
The Physiological Society
Hodgkin Huxley House
30 Farringdon Lane
London, EC1R 3AW
UK
<http://www.physoc.org>
<http://journals.physoc.org>

REQUIRED ITEMS:

-Please include an Abstract Figure. The Abstract Figure is a piece of artwork designed to give readers an immediate understanding of the research and should summarise the main conclusions. If possible, the image should be easily 'readable' from left to right or top to bottom. It should show the physiological relevance of the manuscript so readers can assess the importance and content of its findings. Abstract Figures should not merely recapitulate other figures in the manuscript. Please try to keep the diagram as simple as possible and without superfluous information that may distract from the main conclusion(s). Abstract Figures must be provided by authors no later than the revised manuscript stage and should be uploaded as a separate file during online submission labelled as File Type 'Abstract Figure'. Please ensure that you include the figure legend in the main article file. All Abstract Figures should be created using BioRender. Authors should use The Journal's premium BioRender account to export high-resolution images. Details on how to use and access the premium account are included as part of this email.

EDITOR COMMENTS

Reviewing Editor:

The revised MS is significantly improved and now provides the likely molecular identity responsible for the background Ca influx. Both reviewers recommended acceptance and I concur. However, before formal acceptance, please provide an Abstract Figure (and legend) in a revised version (as per JP policy).

REFEREE COMMENTS

Referee #1:

The authors favourably addressed all my comments. I have no further suggestion and I would like to congratulate the authors to such an interesting manuscript.

Referee #2:

My concerns were fully addressed, no further questions.

Thank you.

END OF COMMENTS

1st Confidential Review

04-Feb-2022

PROFESSOR ANDREW W TRAFFORD
Unit of Cardiac Physiology
University of Manchester
3.24 Core Technology Facility
46 Grafton Street
Manchester
M13 9NT

Email: Andrew.W.Trafford@manchester.ac.uk

Phone: +44-161-275-7969

Fax: +44-161-275-2703

Dr Kim Barrett
Editor-in-Chief
The Journal of Physiology

20th February 2022

Dear Dr Barrett,

RE: Original article, JP-RP-2022-282168X 'Interaction of background Ca²⁺ influx and sarcoplasmic reticulum threshold determine the propensity to Ca²⁺ waves and effects of heart failure in sheep heart.'

On behalf of all of the co-authors I would like to thank you for your decision letter of 10th February 2022. In line with your recommendations, we are pleased to enclose a graphical abstract.

As a final point of note, we have made a final change to the text (the final sentence of the discussion), where we comment that a future area of research will be in understanding the contribution of the background influx to adverse cardiac remodeling. We enclose a marked up copy with these tracked changes.

Once again, we thank the editorial team and reviewers for their careful reviews and are delighted that our manuscript is considered acceptable for publication in the *Journal of Physiology*.

Yours Sincerely,

Prof. A.W. Trafford
Chair of Cardiac Pathophysiology

Dear Dr Trafford,

Re: JP-RP-2022-282168XR1 "Interaction of background Ca²⁺ influx, sarcoplasmic reticulum threshold and heart failure in determining propensity for Ca²⁺ waves in sheep heart" by David Hutchings, George Madders, Barbara Niort, Elizabeth Bode, Caitlin Waddell, Lori Woods, Katharine Dibb, David A Eisner, and Andrew W Trafford

I am pleased to tell you that your paper has been accepted for publication in The Journal of Physiology.

NEW POLICY: In order to improve the transparency of its peer review process The Journal of Physiology publishes online as supporting information the peer review history of all articles accepted for publication. Readers will have access to decision letters, including all Editors' comments and referee reports, for each version of the manuscript and any author responses to peer review comments. Referees can decide whether or not they wish to be named on the peer review history document.

Are you on Twitter? Once your paper is online, why not share your achievement with your followers. Please tag The Journal (@jphysiol) in any tweets and we will share your accepted paper with our 23,000+ followers!

The last Word version of the paper submitted will be used by the Production Editors to prepare your proof. When this is ready you will receive an email containing a link to Wiley's Online Proofing System. The proof should be checked and corrected as quickly as possible.

Authors should note that it is too late at this point to offer corrections prior to proofing. The accepted version will be published online, ahead of the copy edited and typeset version being made available. Major corrections at proof stage, such as changes to figures, will be referred to the Reviewing Editor for approval before they can be incorporated. Only minor changes, such as to style and consistency, should be made a proof stage. Changes that need to be made after proof stage will usually require a formal correction notice.

All queries at proof stage should be sent to TJP@wiley.com

Yours sincerely,

Professor Don M. Bers
Senior Editor
The Journal of Physiology
<https://jp.msubmit.net>
<http://jp.physoc.org>
The Physiological Society
Hodgkin Huxley House
30 Farringdon Lane
London, EC1R 3AW
UK
<http://www.physoc.org>
<http://journals.physoc.org>

P.S. - You can help your research get the attention it deserves! Check out Wiley's free Promotion Guide for best-practice recommendations for promoting your work at www.wileyauthors.com/eeo/guide. And learn more about Wiley Editing Services which offers professional video, design, and writing services to create shareable video abstracts, infographics, conference posters, lay summaries, and research news stories for your research at www.wileyauthors.com/eeo/promotion.

*** IMPORTANT NOTICE ABOUT OPEN ACCESS ***

Information about Open Access policies can be found here <https://physoc.onlinelibrary.wiley.com/hub/access-policies>

To assist authors whose funding agencies mandate public access to published research findings sooner than 12 months after publication The Journal of Physiology allows authors to pay an open access (OA) fee to have their papers made freely available immediately on publication.

You will receive an email from Wiley with details on how to register or log-in to Wiley Authors Services where you will be able to place an OnlineOpen order.

You can check if your funder or institution has a Wiley Open Access Account here <https://authorservices.wiley.com/author-resources/Journal-Authors/licensing-and-open-access/open-access/author-compliance-tool.html>

Your article will be made Open Access upon publication, or as soon as payment is received.

If you wish to put your paper on an OA website such as PMC or UKPMC or your institutional repository within 12 months of publication you must pay the open access fee, which covers the cost of publication.

OnlineOpen articles are deposited in PubMed Central (PMC) and PMC mirror sites. Authors of OnlineOpen articles are permitted to post the final, published PDF of their article on a website, institutional repository, or other free public server, immediately on publication.

Note to NIH-funded authors: The Journal of Physiology is published on PMC 12 months after publication, NIH-funded authors DO NOT NEED to pay to publish and DO NOT NEED to post their accepted papers on PMC.

EDITOR COMMENTS

Reviewing Editor:

Excellent work!

2nd Confidential Review

21-Feb-2022